# A Multifaceted Educational Intervention in the Doctor–Patient Relationship for Medical Students to Incorporate Patient Agendas in Simulated Encounters

**DOI:** 10.3390/healthcare11121699

**Published:** 2023-06-09

**Authors:** Sophia Denizon Arranz, Diana Monge Martín, Fernando Caballero Martínez, Fernando Neria Serrano, Patricia Chica Martínez, Roger Ruiz Moral

**Affiliations:** 1Faculty of Medicine, Universidad Francisco de Vitoria, 28223 Madrid, Spain; 2IMIBIC (Instituto Maimónides de Investigación Biomédica de Córdoba), 14004 Córdoba, Spain

**Keywords:** medical students, communication skills, doctor–patient relationship, medical education, simulated patients

## Abstract

From the beginning of their clinical training, medical students demonstrate difficulties when incorporating patient perspectives. This study aimed to assess if students, after an instructional programme, increased their sensitivity towards patients’ needs and carried out bidirectional conversations. An observational study involving 109 medical students prior to their clerkships was designed. They attended a five-step training programme designed to encourage the use of communication skills (CSs) to obtain patients’ perspectives. The course developed experiential and reflective educational strategies. The students improved their use of CSs throughout three sessions, and the overall score for these patient consultations went up in the opinions of both the external observer (EO) (5; 6.6; 7.5) and the simulated patients SPs (5.3; 6.6; 7.8). Most of the students (83.9%) considered that the CSs addressed were useful for clinical practice, particularly the interviews and the feedback received by the SP and the lecturer. The programme seems to help the students use CSs that facilitate a more bidirectional conversation in a simulated learning environment. It is feasible to integrate these skills into a broader training programme. More research is needed to assess whether the results are applicable to students in real settings and whether they influence additional outcomes.

## 1. Introduction

Teaching students how to handle encounters with patients is a priority objective of medical education that is included in the syllabus through the teaching of communication skills [1,2,3]. This teaching is conceptually framed within a model of so-called “patient-centred” care, which has an ethical and pragmatic basis [4,5], and addresses the need to incorporate and deal with clinical information that is both tangible (empirical) and intangible (hermeneutic) [6].

“Patient centered” approaches are mainly founded on pioneering works, such as those of Pedro Laín Entralgo [7], George Engel [8], or Carl Rogers [9]. A significant institutional support for these approaches in the current health systems in the field of clinical practice is the IOM recommendations on the standards for quality clinical practice [10].

Patient-centred approaches are characterised by the doctor’s emphasis on developing an empathetic, validating, and open attitude towards the patient, incorporating a holistic health perspective that considers the patient’s different biographical dimensions, including their family, work, and sociocultural situation [11]. This requires the doctor to have conversations in which they explore, grasp, and acknowledge the patient’s needs, expectations, and desires (their “agendas”) [12], which are often not explicitly expressed by patients [13].

However, students apparently find these communication skills particularly difficult to acquire and maintain. Although they appreciate the conceptual value of a patient-centred approach, students have problems when it comes to addressing and exploring patients’ life situations in greater depth [14,15]. Especially in the early stages of their training, when their understanding of the functional relationships between psychosocial aspects and health problems are still incomplete, students are required to learn a structured way of incorporating the more technical aspects of clinical questioning (medical history) and physical exams [16]. In addition, they encounter conversational difficulties in such basic aspects as, for example, responding appropriately to changes in direction or interruptions in the conversation, needing to overcome presuppositions, prejudices, and avoiding parallel inner dialogues that are not in line with the main subject of the interview [17,18,19]. They even display problems connecting with their patients’ life experiences [20,21,22] or ending conversations by manifesting genuine, empathic understanding [22,23]. Many of them feel that they do not deserve their patients’ trust when discussing what they consider to be more intimate aspects, and they are, therefore, very insecure about seeking and obtaining this type of information [14,15,24]. It is evident that students tend to consider medical knowledge as the core of their professional role and the aspect that most legitimises their exploration of patients’ conditions [24]. As such, they favour conversations that are more disease- or doctor-centred in their approach. In the long run, this can lead to a negative attitude towards these psychosocial aspects or to them being considered of lesser importance, even though their interaction in the health-disease process is well-documented in the field of neuroscience [14,25,26,27]. Educational programmes should integrate training on the role of the psychosocial aspects of illness and provide students with practical strategies for exploring and addressing patient agendas, especially in phases where students are introduced to the clinical method, which in our educational system takes place either at the end of pre-clinical courses or at the beginning of clinical training, when students learn to take medical histories and do physical exams. Some years ago, IR McWhinney [28] dubbed this learning approach the patient-centred clinical method and began to teach it, initially in the field of general/family medicine. This kind of educational initiative can help students to gain confidence in these types of conversations, give them the skills they need to focus on their patients as individuals, and help them gain a better understanding of the importance of patient-centred attitudes in clinical practice.

This study, therefore, presents an educational proposition that aims to test medical students’ acceptance of these conversational approaches with patients and to assess the extent to which they increase both their sensitivity to patients and their skills for conducting two-way conversations by integrating and addressing patient agendas.

## 2. Methods

### 2.1. Relevance of the Intervention Programme

Our communication training itinerary spans the third to sixth years of the medical degree and is integrated into Clinical Methods I (CM I), Clinical Methods II (CM II), Clinical Stays I (CSI), and Clinical Stays II (CSII). The method and intervention programme implemented in the third year were previously reported [19,29]. To design the specific intervention programme for the fourth year (CM II), we considered both the main communication difficulties manifested by students in sessions with simulated patients (SP) in CM I (third year) and the practical experiences of students during their rotations in health centres. The main objective of this initiative was to raise the students’ awareness of the importance of patient agendas (the relationship in the health-illness process between patients’ psychosocial and experiential aspects) and to equip them with communication skills (CSs) so that they could build a therapeutic relationship based on trust and confidence. To this end, we worked on the approach to the patients’ emotions as well as those concerns that arise for the interviewer during the consultation. The main core of the educational programme was based on developing a communicative strategy for consultations, structured into 5 steps or communicative dimensions [30]:

*Step 1*—“*Stop*”: Student self-regulates and chooses how to respond at a particular moment. The aim is for the learner to avoid reactive behaviour by being proactive, listening before defending, or justifying themselves.

*Step 2*—“*Empathise through validation*”: Validation legitimises the other person’s perspective without necessarily implying agreement. It aims to encourage the learner to convey their interest and understanding of a patient’s feelings or concerns.

*Step 3*—“*I-message*”: The aim is for the learner, in addition to listening to the patient’s perspective, to be able to -judgmentally but assertively present their own perspective non through non-impositional or instructive communication elements.

*Step 4*—“*Ask*”: The aim is to inquire about the patient’s concerns, beliefs, or resources rather than assuming them.

*Step 5*—“*Agree*”: The student aims to reach a realistic and effective agreement with the patient, taking into account the perspectives of both parties.

The five phases of this strategy are integrated into a structured “CICAA patient-centred” consultation model [31]: (a) elicit information; (b) integrate medical and human perspectives (Exploratory Phase); (c) provide information: help and agree (Resolution Phase).

### 2.2. Design of the Educational Programme

The design of the educational proposition included the following teaching activities (Figure 1):

(a) *Eight demonstration and small group work sessions (2 h/session)*: These sessions used dynamic and participatory methods, such as role-playing, to address the fundamentals and applicability of the patient-centred method. They emphasised the holistic approach to health and the need to consider the psychosocial well-being of patients from both pragmatic (to achieve health objectives) and ethical perspectives. Specifically, the final two sessions dealt with critical incidents, identified during the practical rotations, to be worked on and resolved by applying the tools learnt in the preceding sessions.

(b) *Three individual videotaped interviews with three SPs*: The students, sequentially, had three meetings with SPs in which they had to apply the communicative strategies learned (the 5-step protocol described above), in the context of a patient-centred interview. These meetings lasted a maximum of 10 min. During these sessions, the SPs, who presented typical health problems, at a certain point displayed in a sufficiently explicit way (*open clue*) [32] some kind of personal need, idea, or specific expectation (sometimes discrepant or incongruent), with the aim of prompting the learner to switch to the psychosocial aspects and the use of taught communicative skills. Although each session involved different clinical content, the communicative challenges were similar. The following are examples of the core content of some of these conversations: *patient with dyslipidaemia who refuses to go on a diet and demands direct treatment with a drug when it is not indicated; patient who comes requesting a complementary test that is unwarranted; patient recently diagnosed with type II diabetes who is reluctant to accept this diagnosis; patient who requests an apparently unnecessary home visit; patient who is distressed and sad because their partner has been referred to palliative care*.

(c) *Feedback from the SP to each student*: Immediately after the end of the interview, the SP went back into the consultation room to give the student feedback according to agreed guidelines. They had previously received training in the pre-established protocols as well as specific instructions on how to provide both the information in the consultation and the subsequent feedback [33].

(d) *Video screening and self-evaluation*: After the interview and subsequent personalised feedback from the SP, the students watched the video of their meeting and performed a self-evaluation, reflectively analysing their performance in each of the five communication dimensions to be worked on (the 5 steps).

(e) *Individualised written feedback on the interview from their lecturer*: All students received written feedback on their interview from the lecturer in charge of the programme.

(f) *Discussion and closing workshops*: Finally, small group workshops were held to analyse and discuss the experiences and problems experienced by the students in the interviews in terms of achieving the set objectives.

### 2.3. Participants

This study was carried out as part of the training in clinical communication in the fourth year CM II course and was aimed at students enrolled in the 2020–2021 academic year (*n* = 109).

### 2.4. Measurements

This study involved a triple assessment by an External Observer (EO), the SPs, and the students themselves using specific tools and scales. These questionnaires, used to measure whether the student completed the stages of the communication program, was carried out based on an interview model centred on the patient and the skills to be measured among five members of the teaching team (face validity). The EO was always the same lecturer, with expertise in communication, who actively participated in both the design and implementation of the educational programme.

To assess the reliability of the measurements, a test–retest was carried out on 30 of the sessions obtained with an interval of 2 weeks between the two measurements.

The SP encounters were evaluated by the EO using a Likert scale (from 0 to 3 points), which measured the degree to which students applied the communicative tools during the sessions (0 being the lowest and 3 the highest), according to the 5 established steps (Table 1). In addition, by means of an additional item (from 0 to 10 points), the EO assessed the degree to which the session was considered generally satisfactory (overall impression).

The SPs evaluated the same item (overall impression), giving their impression of the interest shown by the interviewer (the student) in their viewpoints and emotions (Likert scale from 0 to 10 points). Finally, using an electronic online questionnaire, the students’ opinions on the overall teaching programme (satisfaction) and the usefulness of the tested strategies (Likert scale from 0 to 5 points: from 0—“not useful at all” to 5—”totally Useful”) were recorded.

### 2.5. Analysis

The qualitative variables were described using the relative frequency in percentage and count, and the quantitative variables were described using the mean as the centralisation measure as well as the standard deviation for dispersion measure.

To study the reliability of the EO, the test–retest was conducted to assess the difference between two different assessments separated by 2 weeks for the same group of interviews (*n* = 30). The Intraclass Correlation Coefficient and Wilcoxon test were calculated, and no significant differences were observed between the two evaluations made in the 30 interviews (*p* > 0.05) for any of the analysed items. The intraclass correlation coefficient was above 0.94 in all cases, pointing to excellent agreement in the ratings of the two assessment stages.

Comparisons were made between the three interviews for each of the 5 dimensions and differences, which were assessed using a Friedman Test (for no normal variables). Kendall’s W index was used as a measure of effect size, which is usually interpreted as weak (0.1 ≤ 0.3), moderate (0.3 ≤ 0.5), or strong (≥0.5). In addition, pairwise comparisons across time measures were estimated using the Durbin–Conover Test (Holm–Bonferroni correction) for each dimension.

To evaluate the influence of both raters and overtime sessions on overall impression scores, a two-way repeated measures ANOVA analysis was performed.

Additionally, each student was classified as above or below average within each session. The Student’s *t*-test was used to analyse the differences between the overall impressions scores between the EO and SP in above- and below-average student samples. Analyses were carried out using R v4.1 and SPSS V.26 software.

## 3. Results

All 109 fourth-year students conducted the three sessions. Table 2 shows the scores for each of the five communicative dimensions over the three interviews assessed by the OE. In all of them, a significant positive evolution was observed (e.g., the “*Stop*” dimension: 1st interview: mean 1, SD 0.4; 2nd interview: mean 2, SD 0.6; 3rd interview: mean 2.4, SD 0.5 points, (*p* < 0.001)). As can also be seen in Table 2, the steps for which the students attained the highest changes in achievement and improved application from the first to the third interview were the first step (“*Stop*”) and the third step (“*I-Message*”), which reflect the low reactivity of the students when conducting the interview (Kendall’s W coefficient 0.767) and their ability to state their own perspective (Kendall’s W coefficient 0.538). Finally, a significant change in the students’ performance was observed for all dimensions between the first and both the second and third interviews.

Later, the overall impression of both raters (the EO and the SPs) and over time (sessions) was evaluated by a two-way ANOVA analysis. Comparing the mean overall impression scores (0–10 point scale) given to the students in each interview, an improvement over time (*p* < 0.001) similar to that obtained in the previous five dimensions was observed for all 109 students (EO: 1st session: mean 5, SD 1.1; 2nd session: mean 6.6, SD 0.9; 3rd session: mean 7.5, SD 1.0 points, (*p* < 0.001); SP: 1st session: mean 5.3, SD 1.2; 2nd session: mean 6.6, SD 1.1; 3rd session: mean 7.8, SD 1.4 points) (Table 3).

As an additional finding, in general, the SPs gave slightly but not significantly higher scores than the EO (*p* = 0.052, Table 3).

To further determine if there are differences between raters, we analysed the results by dividing the students according to how well they performed in the sessions. It was observed that students with below-average scores obtained higher evaluations from the SPs than from the EO in all the evaluated sessions (*p* < 0.05, Table 4). These differences between raters were not observed for students with above-average scores.

Finally, the satisfaction and usefulness survey (measured as the 4—“very useful” and 5—“totally useful” percentage) was completed by 62 students (participation rate of 57%). Of all the teaching activities carried out in the course, the students perceived the sessions and feedback from the SPs and the comments on the video of the consultation made by the lecturer as being the most useful (90% and 79.1%, respectively). Furthermore, 83.9% considered that the CSs they learned were very useful for their future as doctors (Figure 2).

## 4. Discussion

The proposed educational programme is part of the continuum of a cross-cutting clinical communication syllabus that is being developed throughout the entire medical curriculum [34], which uses experiential methodologies applied in simulated and real-life conditions, such as structured feedback, repetition, reflection, and small group discussion, in line with the principles of “deliberate practice” [35]. During the fourth course in particular, the aim was to raise the medical students’ awareness and provide CSs to be applied in “patient-centred” programmes [6]. The educational evidence highlights the importance of teaching these kinds of transversal competencies in a continuous rather than an isolated manner [36,37] as well as the use of experiential teaching techniques [38,39]. The results obtained from this programme also support its effectiveness from a double perspective: the observation of the CSs employed by the students and the improvement seen in the aspects of the relationship as perceived by the SPs themselves in the interviews and the usefulness as perceived by the students.

During the third year, our students were trained in how to take a medical history and perform a physical exam. In these first encounters, although there were no apparent communication difficulties, the students did experience their first problems in terms of having productive conversations with patients [19,29]. The subsequent course aimed to go a step further in their CSs training, by confronting them with sessions in which the patients clearly presented unexpected proposals. The rationale for such a programme is based on the need, on the one hand, to show how important these proposals are as a reflection of patient agendas and, on the other hand, to minimise the difficulty that doctors, particularly student doctors, have in recognising subtle psychosocial cues [13,15,22,40]. Accordingly, our students were required to minimally explore these cues so that they could genuinely demonstrate their understanding to patients. This required the students to make minimal inquiries into the patients’ life circumstances, to determine what triggered those feelings and what justified the patients’ request or reaction [41]. The aim was, thus, to complete the *empathic engagement cycle* [42]. This study also aimed to encourage students to make an effort to find common ground with patients when dealing with the encountered situations. The results show that the students’ use of these communicative strategies progressed over the three sessions, in line with the positive perception by the SPs. This progress, however, does not seem to be uniform. Although these are still only small differences or trends, it is interesting to note that tasks such as showing empathy or seeking agreement are more difficult to implement than stopping and asking questions, which is consistent with the difficulty that each of the tasks entails and is in line with the findings of other authors who have observed, like us, that students have greater difficulty in showing empathy or seeking agreement (steps 2 and 5) than in listening and asking certain probing questions (steps 1 and 4) [21,22,43]. This study does not allow us to assess to what extent students successfully perform these skills in real clinical practice.

Communication has a double component, a behavioural one determined by the implementation of concrete CSs and another one based on the intangible aspects experienced by its stakeholders, the SPs: the “being in relation” [44,45,46]. This study was intended to reflect the impact of these two perspectives, which are usually not fully coincident [47,48]. SPs are an adequate source of information on the “relational state” perspective [48], and their evaluations revealed differences with respect to the external evaluations [49,50]. In our study, the evaluations of the SPs and the EO were fairly well-aligned. However, the SPs tended to evaluate students with below-average performance slightly more favourably. Tamblyn et al. [51] found that the less accurate the student’s portrayal or their own performance was, the more lenient the SPs tended to be when grading students. This was related to fatigue, a decrease in role accuracy by the SP and, in the case of students, to underperformance, which could lead to a bias that prevents them from suffering an evaluative disadvantage.

The educational strategies included in our training proposal not only contained a “skill-centred” and exclusively behavioural approach to applying verbal and non-verbal CSs but also emphasised the experiential aspects. This is achieved by incorporating other educational strategies [52] such as “priming” (students reporting on their own mental process), “reflective questioning” (to unlock possibilities and invite curiosity), and “modelling while thinking aloud” (making the process mentally more transparent: the student asks “*What am I doing and why am I doing it right now?*”) The intention was that, through these activities, students should experience, reflect on, and become aware of these inherent but “intangible” aspects of illness and healthcare.

It is difficult to assess the extent to which this has been achieved. However, the perceptions of the encounters offered by the SPs and the opinions of the students do represent the first, albeit indirect and incomplete, indication of the effectiveness of these aspects or dimensions.

This study has other limitations that need to be pointed out. As mentioned above, this educational programme forms part of a communication curriculum. Ideally, the effectiveness of such a curriculum should be tested as a whole at its completion (after the students have finished their studies) and preferably under real clinical conditions. However, it is equally important that the innovations included in the curriculum have a well-defined structure and specific objectives that can be analysed in their own right. In various studies, we reported the results obtained in previous programmes with different teaching objectives, to which we now add the results presented here [19,22,27,29,34,40]. Taken together, they can shed light on the possible scope of the programme as a whole, although overall assessments are needed to determine its true impact when these students conduct consultations with real patients. However, the approximations made here are also limited by other aspects, such as the limited number of CSs observed, and the fact that the students’ perspectives were only gathered as impressions on partial aspects of the programme via a survey involving closed questions. It would be interesting to include more targeted open-ended questions and even to carry out other types of qualitative studies (interviews or focus groups) with students to explore the specific aspects of this impact in greater depth. The number of students who completed the satisfaction survey was slightly more than half, and we do not have data on those who did not fill it out, so these results should be taken with caution. Finally, this study was carried out using a homogeneous group from a single institution, and its non-experimental design does not allow conclusions to be drawn about its true effectiveness.

## 5. Conclusions

A communication training programme developed in a simulated environment with an expert observer (EO) and simulated patients (SPs), based on the teaching of both behavioural skills and attitudinal elements, seems to be effective in helping students to conduct more bidirectional interviews with patients. The five-step communication protocol is a simple and practical tool for helping trainees to effectively integrate communication competence and deal with emotions in a clinical consultation. It is feasible to implement this as part of the broader training programme within our usual training syllabus. Further research is needed to assess whether the results are applicable to students at more advanced educational levels or in real-world settings and whether they influence additional outcomes.

## Figures and Tables

**Figure 1 healthcare-11-01699-f001:**
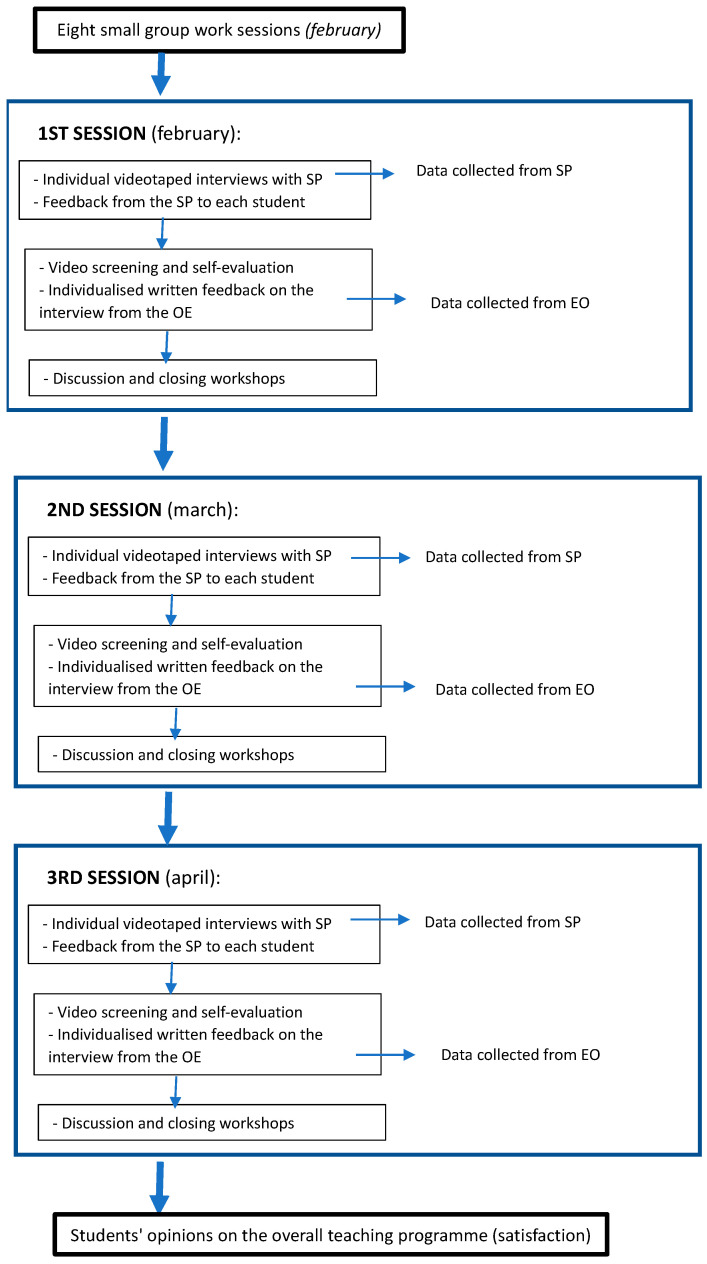
Programme flowchart.

**Figure 2 healthcare-11-01699-f002:**
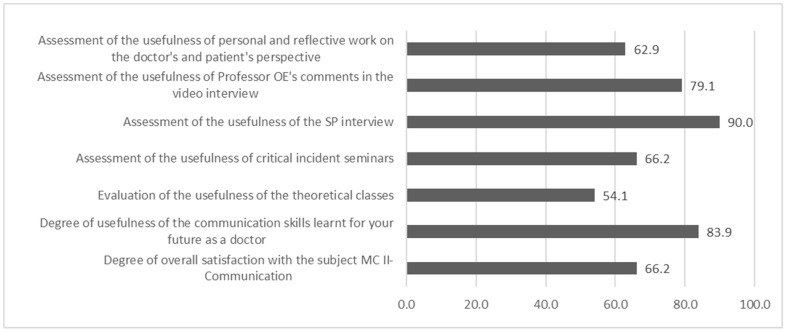
Student evaluation of the teaching activities carried out in the subject (4—“very useful” and 5—“totally useful” percentage) *n* = 62.

**Table 1 healthcare-11-01699-t001:** Dimensions assessed in the interviews (by the EO).

Step 1 “*Stop*”: Responds with low reactivity, responding to the patient’s emotion/confrontation; Step 2 “*Validate*”: Shows empathy; Step 3 “*I-Message*”: States own perspective or informs; Step 4 “*Ask*”: Inquires about the patient’s perspective; Step 5 “*Agreement*”: Tries to reach an agreement without imposing;
Overall impression of the degree to which the student focuses on the patient in the consultation.

**Table 2 healthcare-11-01699-t002:** Mean and median scores for each of the items in the first, second, and third sessions (*n* = 109).

	1st Interview ^1^	2nd Interview ^1^	3rd Interview ^1^	*p*-Value ^2^	Kendall’s W Coefficient ^3^	Effect Size Magnitude	1st vs. 2nd *p*-Value ^4^	2nd vs. 3rd *p*-Value ^4^	1st vs. 3rd *p*-Value ^4^
Stop	1.0 (0.4)	2.0 (0.6)	2.4 (0.5)	**<0.001**	0.767	large	**<0.001**	**<0.001**	**<0.001**
Validate	1.3 (0.6)	1.9 (0.5)	2.1 (0.6)	**<0.001**	0.348	moderate	**<0.001**	0.102	**<0.001**
I-Message	1.1 (0.6)	2.0 (0.6)	2.2 (0.4)	**<0.001**	0.538	large	**<0.001**	0.05	**<0.001**
Ask	1.3 (0.6)	2.1 (0.5)	2.3 (0.6)	**<0.001**	0.463	moderate	**<0.001**	**0.028**	**<0.001**
Agree	1.4 (0.5)	1.9 (0.4)	2.0 (0.4)	**<0.001**	0.474	moderate	**<0.001**	0.079	**<0.001**

^1^ Mean (SD); ^2^ Friedman test; ^3^ effect size; ^4^ Durbin–Conover (corrected Holm–Bonferroni) post hoc comparison test.

**Table 3 healthcare-11-01699-t003:** Overall impression scores of the EO and SP in the three sessions (*n* = 109).

	1st Session ^1^	2nd Session ^1^	3rd Session ^1^	*p*-Value ^2^ (between Sessions)	*p*-Value ^3^ (between Raters)
External Observer	5.0 (1.1)	6.6 (0.9)	7.5 (1.0)	**<0.001**	0.052
Simulated Patient	5.3 (1.2)	6.6 (1.1)	7.8 (1.4)		

^1^ Mean (SD); ^2^ two-way ANOVA (between sessions); ^3^ two-way ANOVA (between raters).

**Table 4 healthcare-11-01699-t004:** EO and SP overall impression scores from interviews as a function of below-average and above-average student performance.

	1st Session ^1^		2nd Session ^1^		3rd Session ^1^	
BELOW AVERAGE	*n* = 74	*p*-value ^2^	*n* = 48	*p*-value ^2^	*n* = 59	*p*-value ^2^
External Observer	4.4 (0.8)	**0.006**	5.8 (0.4)	**0.034**	6.7 (0.4)	**<0.001**
Simulated Patient	4.8 (1.0)		6.1 (1.0)		7.5 (1.4)	
ABOVE AVERAGE	*n* = 35	*p*-value ^2^	*n* = 61	*p*-value ^2^	*n* = 50	*p*-value ^2^
External Observer	6.3 (0.5)	0.372	7.3 (0.5)	0.078	8.5 (0.5)	0.159
Simulated Patient	6.2 (0.9)		7.0 (1.1)		8.2 (1.3)	

^1^ Mean (SD). ^2^ Welch’s two-sample *t*-test.

## Data Availability

The data that support the findings of this study are available on request from the corresponding author.

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
