# Peer review of "A Multifaceted Educational Intervention in the Doctor–Patient Relationship for Medical Students to Incorporate Patient Agendas in Simulated Encounters"

_healthcare, 2023, doi:10.3390/healthcare11121699_

Round 1

Reviewer 1 Report (Previous Reviewer 1)

1) The theoretical framework still needs to be developed.

2) Still no flowchart/description of the study over time. What was the distribution of training sessions and measurements taken on the timeline?

3) EO and SP scores continue to be treated as measurements on the same scale. Table 3. shows the results of the 2-way ANOVA, and Table 4. shows the results of the Welch Two Sample t-test. Both analyses assume that the measurement is on the same interval scale and we are comparing means. This is a risky model for this study, especially since the EOs and SPs used 10-point scales with different criteria (the EOs assessed the degree to which the session was considered generally satisfactory vs SPs the SPs evaluated in terms of the interest shown by the interviewer) In the case of the analysis in Table 3. it is not clear whether this is an ANOVA for repeated measurements. In general, evaluation of training effectiveness does not require direct comparison of EO and SP ratings. Trends over time are important.

4) It is still difficult to understand the reliability assessment (test-retest) for EO estimates. How was the ICC applied? What do values above 0.94 mean? Why weren't test-retest correlations (e.g., correlations for rank data or Pearson's r) or measures of absolute consistency of assessments (e.g., normalized sums of t1 and t2 differences) counted?

5) The analyses presented in Table 4. are unnecessary, not related to the problem of training effectiveness.

Author Response

Reviewer 2 Report (New Reviewer)

Dear Authors.

After reviewing the document, I have the following comments and considerations to make:

- The abstract seems adequate to me, although it should be specified what the abbreviations used refer to.

- The introduction of this article is well elaborated, adequately addressing the existing problems of the research topic.

- The established objectives are adequate to clarify the study problem.

- The methodology allows adequately addressing the study problem and achieving the proposed objectives, although I consider that it should be specified if in the test-retest the same problem has been addressed with the same simulated patient, because if this is not the case it would not be useful since both tests are not comparable (lines 185-187).

- The results have been presented in a clear manner to facilitate their understanding.

- The discussion presented in this research provides a detailed and deep analysis of the results obtained, establishing relationships between the findings and previous studies existing in the scientific literature.

- The conclusions presented in this research are coherent with the results obtained and respond adequately to the proposed objectives.

Kind regards.

Author Response

Thank you very much for your comments.

Reviewer 3 Report (New Reviewer)

First, this is focuses on a critical clinical set of skills.

This instruction appears to align with the biopsychosocial approach to medicine, but I don't see a citation to that approach. 

It is unclear in the methods what is the nature of the agreement. Is it an agreement about perspectives or is it about a plan of treatment? Either way, what guidance is the student provided about how to navigate an agreement if the perspectives are inconsistent?

I would have appreciated more detail about the small group work. Do all students watch each other? The description of the disparity between SP and student evaluation was interesting. Is it possible that the SPs have some intuition about the fragility of the students who overrate their performance and, as a result, overrate their performance as well, to soften the critique? This makes me wonder about how the SPs were trained in evaluation. 

Round 2

Reviewer 1 Report (Previous Reviewer 1)

Responses to critical comments are perfunctory and do not clarify all concerns. E.g. the development of the theory of intervention is insufficient, the justification that the assessments of EOs SPs are invariant is unconvincing, the argument that unrelated to the topic of the article the analyses will not be removed because they are interesting according to the Authors is also difficult to accept. However, due to the importance of the study, I suggest the article be published.

Reviewer 3 Report (New Reviewer)

The authors adequately addressed the issues identified in the first review.

This manuscript is a resubmission of an earlier submission. The following is a list of the peer review reports and author responses from that submission.

Round 1

Reviewer 1 Report

A very important topic, both theoretically and practically, but the study itself and its description raises many questions.

1) Lack of reference to psychological theories of taking the perspective of another (older ones: e.g. Feffer's theory of interpersonal decentration, newer ones: construct of theories of mind) and related lack of a broader theoretical underpinning of the planned intervention. 

2) IIntervention contaminates with measurement - lack of information about the ordering of intervention components on the timeline. Could the changes between the 1st and 2nd measurement (EO) be related to feedback from the SP, or is it more of an effect of demonstration and small group work sessions, or is individualized written feedback on the interview from their lecturer key? This is all the more important since the results in Table 2. indicate that progress is mainly observed between the 1st and 2nd measurements. 

3)Although in the paper there is statement "Finally, the study (...) non-experimental design does not allow conclusions to be drawn about its true effectiveness." (305-306), it was de facto an experiment, just poorly designed. In the study design a control group has lacked , and it was possible to randomly divide the subjects into two groups and implement time-shifted treatments in them.

4) There are not basic statistics for variables, way of estimating the reliability of assessments by external experts is incomprehensible. Since the analyses involve changes over time, it would be useful to reflect on the invariance of EO assessments over time (e.g., whether EOs in subsequent measurements are equally harsh in their assessments). The descriptions in the tables cause puzzlement: "Mean ± SD." Why plus/minus? Is SD being confused with SE? The tables lack information on abundance. Is it always n=109? No measures of effect strength.

 5) Analysis model for EO scores does not allow to assess dynamics of change (1st vs. 2nd measurement, 2nd vs. 3rd measurement, 1st vs. 3rd measurement). A model with repeated measurements would be appropriate. This is even more important since the Discussion reads "The results show that the students' use of these communicative strategies progressed over the three sessions, in line with positive perceptions by the SPs." (253-254).

6) The analysis model for comparisons of EO and SP assessments is problematic. The t-test model for dependent samples requires that the dependent variable be measured on the same scale. EO and SP ratings certainly do not meet this condition (in the case of estimation scales, the rater must be treated as the defining factor of the scale). A t-test model for independent samples and treating the results as an analysis of the rater effect would be more appropriate.

7) For the reliability of the evaluation of the effectiveness of the experiment, the low percentage of student participation in the evaluation of usefulness is troublesome.

Reviewer 2 Report

Thank you very much for allowing me to review this article on an educational proposal to improve the therapeutic relationship with patients. It is a study with many possibilities but it has some shortcomings that I list below with the intention of contributing to improve the quality of the study.

Firstly, I have missed the approval of an ethics committee. The authors indicate that it is not necessary because it is part of the curriculum but these data have been used for research purposes and will be used in a publication.

In terms of methodology, there is no indication of how the sample was drawn, nor is there any reference to the population size, nor are there any socio-demographic data to help the reader situate the context in which the research was carried out.

The questionnaire used to measure whether the student carried out the stages of the communication programme is not a validated questionnaire, it has been developed by the authors to measure educational activity. The first step could be to check the reliability and validity of the measuring instrument. 

The results obtained indicate that the more the activity is repeated, the better the results are obtained. However, it is not possible to know whether their attitude or predisposition to deal with the patient from a position of equality has changed with the repetition of the simulation sessions.

As for the survey on satisfaction with the activity, it was completed by approximately 50% of the initial participants in the activity, so it does not have a representative value. 

I believe that it is very important to carry out these interventions with doctors in order to improve the relationship they establish with patients. I encourage the authors to validate the questionnaire used and to use other validated questionnaires to measure whether changes in the attitudes of the students are produced that can subsequently be reflected in the relationship established with patients.